# Targetable Pathways in Advanced Bladder Cancer: FGFR Signaling

**DOI:** 10.3390/cancers13194891

**Published:** 2021-09-29

**Authors:** Jin-Fen Xiao, Andrew W. Caliri, Jason E. Duex, Dan Theodorescu

**Affiliations:** 1Division of Medical Oncology, Cedars-Sinai Medical Center, Los Angeles, CA 90048, USA; jinfen.xiao@cshs.org (J.-F.X.); andrew.caliri@cshs.org (A.W.C.); 2Department of Surgery (Urology), Cedars-Sinai Medical Center, Los Angeles, CA 90048, USA; 3Cedars-Sinai Samuel Oschin Comprehensive Cancer Institute, Los Angeles, CA 90048, USA; Jason.Duex@cshs.org

**Keywords:** bladder cancer, fibroblast growth factor receptor, mutation, DNA, targeted therapy, kinase inhibitors, erdafitinib

## Abstract

**Simple Summary:**

Around 20% of advanced bladder cancer patients carry unfavorable genetic alterations in the fibroblast growth factor receptor 3 (FGFR3) gene. This review summarizes recent findings from published research and clinical trial data focusing on developing and testing therapeutics that inhibit the increased activities of these genetic alterations. Possible mechanisms of drug resistance observed in some patients are also discussed. This review also discusses clinical findings from studies combining FGFR inhibition with other targeted inhibitors and/or immunotherapy to examine whether outcomes may be improved, especially in patients who have less than optimal responses to FGFR3-directed monotherapy.

**Abstract:**

Bladder cancer is the 10th most commonly diagnosed cancer in the world, accounting for around 573,000 new cases and 213,000 deaths in 2020. The current standard treatment for locally advanced bladder cancer is neoadjuvant cisplatin (NAC)-based chemotherapy followed by cystectomy. The significant progress being made in the genomic and molecular understandings of bladder cancer has uncovered the genetic alterations and signaling pathways that drive bladder cancer progression. These developments have led to a dramatic increase in the evaluation of molecular agents targeting at these alterations. One example is Erdafitinib, a first-in-class FGFR inhibitor being approved as second-line treatment for locally advanced or metastatic urothelial carcinoma with FGFR mutations. Immunotherapy has also been approved as second-line treatment for advanced and metastatic bladder cancer. Preclinical studies suggest targeted therapy combined with immunotherapy has the potential to markedly improve patient outcome. Given the prevalence of FGFR alternations in bladder cancer, here we review recent preclinical and clinical studies on FGFR inhibitors and analyze possible drug resistance mechanisms to these agents. We also discuss FGFR inhibitors in combination with other therapies and its potential to improve outcome.

## 1. Introduction

Bladder cancer is the most common malignant disease in the human urinary system [1]. Once a tumor has invaded into the muscle layer of the bladder, bladder cancer is diagnosed as muscle invasive or advanced bladder cancer. This cancer can present as localized or spread to the lymph nodes or distant organs. Even with adequate treatment localized disease has an elevated risk for systemic spread and threat to life [2]. 

Treatment options for patients diagnosed with advanced bladder cancer have changed recently and are likely to continue changing as advances in our molecular understanding continues [3]. For localized disease, standard of care remains neoadjuvant chemotherapy using a gemcitabine-cisplatin (GC) or methotrexate, vinblastine, doxorubicin plus cisplatin (MVAC) [4] regimens, followed by radical cystectomy [5,6,7,8]. For advanced bladder cancer that has spread, therapeutic options have changed dramatically and in addition to chemotherapy with the same two regimens above, immune checkpoint therapy (ICT) has become a new second-line standard [9].

In addition to ICT, targeted therapies directed at drivers of bladder cancer cell malignancy are being actively developed [10]. Currently identified driver genetic abnormalities and signaling pathways of advanced bladder cancer include mutations in *TERT* gene promoter, *FGFR3* mutations, and *ERBB2* and *ERBB3* mutations. Currently identified driver signaling pathways include FGFR, EGFR, PI3/AKT/mTOR, RAS-MAPK, cell cycle checkpoint, and DNA damage repair [11,12,13]. All these genetic variations are currently being targeted in pre-clinical or clinical investigations with targeted therapies. In April 2019, a clinical trial (ClinicalTrials.gov number, NCT02365597) showed that Erdafitinib, a pan-FGFR inhibitor, achieved a 42% overall response rate in patients with locally advanced or metastatic urothelial carcinoma carrying susceptible FGFR mutations [14]. Erdafitinib was granted accelerated approval as a second-line treatment option for advanced bladder cancer patients. However, 58% of patients who have a susceptible *FGFR3* alteration exhibit no response and among the 42% responsive patients, 39% patients were only partially responsive to Erdafitinib [14]. Importantly, this clinical trial was performed in patients having failed chemotherapy and/or immunotherapy, and thus these treatment outcomes may not be applicable to all bladder cancers with *FGFR3* mutations. Furthermore, the low complete response rate to Erdafitinib raises the possibility that combination therapy with other inhibitors and/or immune checkpoint blockade may improve outcomes. 

In this review, we summarize recent findings on FGF/FGFR signaling, activating mutations, and other alterations of *FGFR3* in bladder cancer. We discuss inhibitors targeting FGFR alterations and on-going trials and possible mechanisms for drug resistance. We also summarize the potential of combination therapies using FGFR inhibitors in clinical trials.

## 2. Overview of FGF/FGFR Signaling

The fibroblast growth factors (FGF)-FGF receptor signaling pathway impacts cell proliferation, differentiation, angiogenesis, metabolism, mobility, and invasion [15,16]. There are five receptors in the FGFR family (FGFR1-4 and FGFRL1). FGFRs are single-pass transmembrane proteins with an extracellular domain, a transmembrane domain, and an intracellular domain. The extracellular domain of FGFRs comprise three immunoglobulin-like domains (IgI, IgII, and IgIII) (Figure 1A) [17]. Except FGFR4, the IgIII domain of FGFR1-3 is subjected to alternative splicing, resulting in IgIIIb and IgIIIc isoforms, which have diverse binding specificities to different FGFs [17,18,19]. The expression of alternatively spliced FGFR transcripts is tissue specific, and essential for the development of some organs. The IIIb splice variants for *Fgfr1* and *Fgfr2* are epithelial tissue specific, while the IIIc splice variants of *Fgfr1* and *Fgfr2* often are expressed in mesenchymal tissue [20,21,22,23]. In contrast, both splice variants of *Fgfr3* are found in epithelial tissue [24,25]. FGFRL1 is a truncated FGFR retaining high-affinity binding of some FGFs yet lacking the intracellular tyrosine kinase domain required for signal transduction [26,27]. FGF binding activates FGFRs, which in turn phosphorylates adaptor proteins and signal through four major downstream cascades: (1) Ras/Raf/MEK-MAPK, (2) PI3K/AKT, (3) PLCγ, and (4) signal transducer and activator of transcription (STAT) [15,16]. 

In humans, 22 FGF ligands have been identified, which can be clustered into three major FGF groups: canonical FGFs, endocrine FGFs, and intracellular FGFs [15] (Figure 1B). Canonical FGFs include five paracrine subfamilies: FGF1, FGF4, FGF7, FGF8, and FGF9 subfamilies, and bind to four tyrosine kinase FGF receptors (FGF1-4) via a high-affinity interaction with co-factor heparin or heparan sulphate, followed by FGFR activation, dimerization, and activation of cytoplasmic signaling transduction pathways [15,28,29]. Conversely, endocrine or hormone-like FGFs (FGF19 subfamily) have low affinity to FGFRs in the presence of heparin/heparan sulphate [30,31], and instead require co-receptors α-and β-Klotho to bind, thereby activating FGFRs to regulate cell growth and metabolism [32,33,34,35,36]. The intracellular FGFs (FGF11 subfamily) are non-secretory FGFs that interact with and regulate voltage-gated sodium channels and other molecules, such as p65 and NF-κB, to regulate neuronal development and function [37,38,39,40,41]. 

## 3. Hyperactivated FGFR3 in Bladder Cancer

Aberrant genetic alterations of FGFRs, including amplification, fusion, and mutation, result in FGFR signaling hyperactivation, which promotes proliferation, metastasis, and drug resistance in cancer cells [42]. *FGFR3* is the most frequently hyperactivated of the FGFRs in bladder cancer, and its genetic alterations are found in around 20% of advanced bladder cancer [10,43,44]. The most common *FGFR3* alterations in advanced bladder cancer are activating missense mutations and in-frame *FGFR3-TACC3* fusions [10]. In the bladder cancer TCGA, activating missense mutations have been reported at S249C (7.9%) and R248C (0.7%) in the extracellular domain; Y373C (2.0%), G370C (1.2%), S371C (0.5%), and G380R (0.5%) in the transmembrane domain; and K650E (0.5%) in the intracellular kinase domain [10] (Figure 1A). The most common missense mutations (S249C and R248C) are APOBEC-signature mutations (C>T and C>G) [10]. The gain-of-function missense mutations in the extracellular and transmembrane domains of FGFR3 lead to ligand-independent dimerization between mutant receptors, whereas mutations in the intracellular kinase domain promote activation of FGFR3 tyrosine kinase activity [45,46]. Interestingly, the missense mutations of *FGFR3* are associated with higher *FGFR3* mRNA and also protein expression in bladder cancer [47,48,49], but the mechanism behind this upregulation is unclear. *FGFR3-TACC3* fusion occurs in 2% of advanced bladder cancer patients [10]. *TACC3* rearrangement to the C-terminal of *FGFR3* leads to an absence of the typical *FGFR3* 3′-untranslated region (3′-UTR), thereby bypassing microRNA regulation. This event leads to increased FGFR3 protein expression and increased FGFR3 pathway activity [50,51]. 

## 4. Targeting Hyperactivated FGFR3 in Advanced Bladder Cancer 

A number of studies have shown that bladder cancer cells harboring *FGFR3* hyperactivating mutations and *FGFR3-TACC3* fusion are responsive to FGFR3 inhibition [52,53,54,55]. So far, strategies include selective tyrosine kinase inhibitors and monoclonal antibodies. Below we highlight agents that have been evaluated specifically in bladder cancer. Many more are in early investigations that may include some patients with bladder cancer in the context of other tumor types. These are not included here.

### 4.1. Small Molecule Tyrosine Kinase Inhibitors

Erdaftinib (JNJ-42756493) has selective and potent inhibition of all four FGFR proteins, leading to antitumor activity [56]. Oral administration achieved clinical response and a manageable safety profile in patients with brain, urothelial, and endometrial cancers [57]. In another clinical trial, erdafitinib showed tumor response in 40% of previously treated patients who had locally advanced and unresectable or metastatic urothelial carcinoma with FGFR alterations [14]. Based on this trial, erdafitinib was granted accelerated FDA approval for second-line treatment. A phase III study is now being carried out to compare the efficacy of erdafitinib against chemotherapy or immunotherapy (pembrolizumab) in advanced bladder cancer patients with FGFR aberrations, whose disease has progressed after previous treatments (ClinicalTrials.gov number, NCT03390504). Another phase III study aims to expand the usage of erdafitinib to advanced cancer patients with FGFR genetic alterations who have exhausted all treatment options (ClinicalTrials.gov number, NCT03825484). A phase II study is testing the efficacy of erdafitinib for non-muscle invasive bladder cancer with *FGFR3* mutations in their trans-urethral resection of bladder tumor (TURBT) or biopsy samples (ClinicalTrials.gov Identifier, NCT04172675).

Infigratinib (BJG398) is an FGFR1–3-selective oral tyrosine kinase inhibitor [58] shown in a phase II study to reduce the size of tumors bearing *FGFR3* alterations and stabilize disease in metastatic urothelial carcinoma patients [59]. Based on these findings, infigratinib was granted FDA approval [60]. A phase III clinical trial is testing this agent in the adjuvant setting following surgery in advanced bladder cancer with susceptible *FGFR3* genetic alterations (ClinicalTrials.gov Identifier, NCT04197986).

AZD4547 selectively targets the FGFR1-3 tyrosine kinases and inhibits tumor growth in an FGFR-driven human tumor xenograft model [61]. In a phase II study carried out in patients with solid tumors with FGFR alterations, AZD4547 exhibited activity towards cancers harboring FGFR activating mutations and fusion but no other alterations [62]. 

Rogaratinib (BAY 1163877) is a potent and selective FGFR1-4 inhibitor that showed anti-tumor activity in FGFR-addicted cell lines of various cancer types [63,64]. Phase I dose-escalation and dose-expansion in patients with advanced cancer having FGFR genetic aberrations showed that rogaratinib was well tolerated and active [65]. Furthermore, a phase II/III clinical trial comparing the efficacy of rogaratinib and chemotherapy showed that rogaratinib had a comparable efficacy with standard chemotherapy and an acceptable safety profile in patients having *FGFR1-3* mRNA overexpression and *FGFR3* genetic aberrations [66]. This study found that rogaratinib tended to be more active in patients with *FGFR3* genetic alterations. 

Pemigatinib (INCB054828), a selective FGFR1-4 inhibitor, suppressed the growth of cancer cells with activating alterations of FGFR1-3 in xenograft models at low oral doses [67]. In a phase II study, Pemigatinib achieved a 36% overall response rate in previously treated advanced cholangiocarcinoma patients with *FGFR2* susceptibility [68]. These results led to accelerated FDA approval. Pemigatinib is currently being tested in a phase II trial for high-risk urothelial cancer patients after radical surgery (ClinicalTrials.gov Identifier, NCT04294277), as well as non-muscle invasive bladder cancer patients with recurrent low- or intermediate-risk tumors (ClinicalTrials.gov Identifier, NCT03914794).

### 4.2. Neutralizing Antibodies and FGF Ligand-Traps Targeting FGFR

Antibodies targeting FGFR inhibitors and traps of the FGF ligand (FGF-traps) can target FGFR signaling. Neutralizing antibodies block the binding of ligands to cognate receptors, thus preventing receptor signaling, and they often perform these functions at low doses, with minimal toxicity. 

Vofatamab or B-701 is a monoclonal antibody that targets the extracellular domain of both wild-type and mutant *FGFR3* and inhibits bladder cancer cell line proliferation [69]. Patients with advanced and metastatic bladder cancer who have failed platinum-based chemotherapy were enrolled in FIERCE-21, a phase Ib/II study, and treated with vofatamab alone or in combination with docetaxel [70]. Vofatamab was well tolerated, and a proportion of patients benefited from vofatamab in combination with docetaxel [70]. In a phase Ib/II trial, vofatamab with the anti-PD-1 checkpoint inhibitor pembrolizumab in patients with platinum-refractory metastatic urothelial was well tolerated [71]. However, there was no correlation of response with *FGFR3* mutations or fusions [71]. 

R3Mab is an antibody that selectively targets the IgII and IgIII domains of both wild-type and mutant FGFR3 receptors to prevent ligand binding and signaling [72]. R3Mab demonstrated robust antitumor activity in FGFR3-dependent tumor cells in xenograft models [72]. However, the extremely high specificity of R3Mab for FGFR3 turned out to be easily circumvented by cancer cells, as the other FGFRs were able to bind ligands and compensate for the reduced FGFR3 activity. Therefore, efforts have been taken to modify the monoclonal antibody and expand the selectivity for FGFR2 and FGFR4, while sparing FGFR1 [73].

FGF-ligand traps are soluble engineered proteins that absorb multiple FGF ligands. FP-1039 and sFGFR3 are FGF traps that contain extracellular domains of FGFR1 and FGFR3, respectively, which allows them to bind and sequester FGF ligands [74,75]. Injection of FP-1039 into mice neutralized FGFR1 ligands (FGF1, FGF2, and FGF4) and inhibited FGFR tyrosine kinase activity [74]. Injecting soluble sFGFR3 into mice harboring tumors with FGFR3 activating mutations neutralized FGFR ligands (FGF2, FGF9, FGF18, etc.) and rescued the symptoms of FGFR3 germline mutation, which causes achondroplasia of mice after birth [75]. 

Currently, neutralizing antibodies and FGF-ligand traps are not under active clinical trial recruitment for advanced bladder cancer patients, but these preclinical results are very encouraging.

## 5. Resistance to FGFR Inhibitors in Cancer

Resistance to FGFR inhibition may happen by three mechanisms: (1) gatekeeper mutations of FGFR; (2) activation of compensatory or parallel signaling pathways; and (3) hyperactivation of downstream stimulators or abolishment of negative regulators. 

Acquiring gatekeeper mutations in the kinase domain, which in turn modulates the ATP binding ability of the kinase, is a common cancer escape mechanism of tumors subjected to small molecule tyrosine kinase inhibitors [76,77]. In FGFR-driven leukemia, the activation mutation V561M within the *FGFR1* tyrosine kinase domain, coupled with inactivation of *PTEN*, led to increased PI3K/AKT activity and acquired resistance to AZD4547 and infigratinib [78]. To overcome the resistance by the FGFR1-V561M mutation, a third generation of the ABL inhibitor GZD824 was developed and showed efficacy in inhibiting the FGFR1-V561M mutant in xenograft tumors [79]. Sequencing of *FGFR3* in a derivative cell line from the myeloid cell line KMS-11 (FGFR3-Y373C) that acquired resistance to FGFR inhibitors revealed a secondary gatekeeper mutation at the kinase domain (*FGFR3-V555M*) [80]. Treating this derivative with an FGFR inhibitor failed to inhibit the kinase activity of this FGFR3, suggesting the kinase cascades were unaffected [80]. 

FGFR signaling pathways are shared by many other tyrosine kinase receptors, such as EGFR, PDGFR, VEGFR, TRK, IGFR, and Tie1,2 (Figure 1D). Thus, “kinase switching” is a known compensatory signaling mechanism in response to FGFR inhibition. A functional screen with bladder cancer cells RT112 (FGFR3-TACC3) showed that the PI3K pathway is the mechanism of resistance to the FGFR inhibitor AZD4547 [81]. Other examples include the activation of EGFR, ERBB3, or PI3K-protein kinase B pathways [81]. Likewise, activation of FGFR signaling rendered EGFR-driven cancer cells resistant to EGFR inhibitors, and this EGFR resistance was overcome by FGFR inhibition [82,83,84]. A CRISPR functional genomic screen revealed that FGFR signaling contributed to the resistance of the EGFR gatekeeper mutation (EGFR-T790M) to EGFR inhibition in EGFR-driven non-small cell lung cancer [85]. Combining EGFR and FGFR inhibitors suppressed the survival and expansion of EGFR-mutation drug-resistance cells [85]. Moreover, ERBB2-overexpressing breast cancer cells gained resistance to HER2 blockade via increasing the FGF3/4/19 copy number and FGFR phosphorylation [86]. 

FGFR inhibitor resistance can involve activating mediators of FGFR signaling or suppression of negative regulators downstream of FGFR signaling. Lung cancer cell lines with *FGFR1* amplification were highly sensitive to FGFR inhibitors [87]. However, these cells exhibited a sustained residual cellular viability due to subclonal existence of drug-resistant cells [87]. These *FGFR1*-amplified lung cancer cells showed primary resistance to the FGFR inhibitors AZD4547 or erdafitinib and were characterized by sustained MAPK pathway activation from constitutive MET and RAS activation, as well as deletion of *DUSP6*, a negative regulator of MAPK signaling [87]. Similarly, KRAS-driven lung and pancreatic cancer cells showing limited response to an MEK inhibitor showed upregulation of FGFR signaling [88]. Suppression of FGFR signaling together with an MEK inhibitor led to regression of KRAS-mutant lung tumors [88]. One of the major contributors to drug resistance in “oncogene-addicted” cancer cells is STAT3 activation [89]. Since the oncogenic drivers EGFR, ERBB2, ALK, MEK, mutant KRAS, and FGFR are all in this STAT3 activation feedback loop, an inhibitor combination strategy may prove effective [89].

## 6. FGFR Inhibitors in Combination Therapies

In advanced bladder cancer, a phase II clinical trial is studying the efficacy and safety of erdafitinib in combination with a sensitive cytochrome 450 (CYP) 3A substrate (midazolam) and with an organic cation transporter 2 (OCT2) probe substrate (metformin).

Inhibition of immune checkpoints with anti-programmed death 1(PD-1) and its ligand (PD-L1) therapies has shown manageable cytotoxicity with durable response in metastatic bladder cancer [90,91]. Advanced bladder cancer with mutations in *FGFR3* are associated with a lower inflammation signature [10]. In a tobacco carcinogen OH-BBN-induced mouse bladder cancer model, the *FGFR3-S249C* mutation led to enhanced bladder tumorigenesis but also suppressed the acute inflammatory response at an early tumor initiation stage [92]. These observations suggest bladder cancers with FGFR3 mutations may have a lower response rate to ICT, as the inflammatory signature is one of the indicators for response to ICT. However, a clinical trial with PD-1/PDL1 blockade in patients with metastatic bladder cancer did not show significant differences in response rates in patients with mutant *FGFR3* versus wild-type *FGFR3* [47]. On the other hand, treating mutant FGFR2-driven lung cancer with erdafitinib showed that inhibition of FGFR signaling increased T cell infiltration, decreased regulatory T cells, and downregulated PD-L1 expression [93]. In addition, erdafitinib in combination with anti-PD-1 lead to increased TCR clonality and decreased tumor-associated macrophages [93]. Another study also showed that inhibiting FGFR with FIIN4, a covalent inhibitor, also increased CD8+ lymphocytes and reduced myeloid suppressor cells in preclinical models [94]. Moreover, FIIN4 in combination with anti-PD-L1 enhanced the survival of mice with pulmonary tumors [94]. Currently, several trials are examining FGFR inhibitors in combination with anti-PD-1/PD-L1 in advanced bladder cancer (Table 1). However, the BISCAY trial employing AZD4547 in combination with anti-PD-L1 in advanced urothelial cancer patients harboring FGFR3-aberrations failed to meet efficacy criteria for further clinical developments [95].

## 7. Conclusions

Targeting *FGFR3* genetic aberrations with pan-FGFR inhibitors has demonstrated clinical benefits in advanced bladder cancer. However, several mechanisms can underlie the low rates of complete response for patients with FGFR alterations. These include in vivo potency, specificity and primary, and acquired resistance. Moreover, the results of the FGFR inhibitor AZD4547 with anti-PD-L1 suggests the need for improved patient selection strategies and/or improved combination regimens.

## Figures and Tables

**Figure 1 cancers-13-04891-f001:**
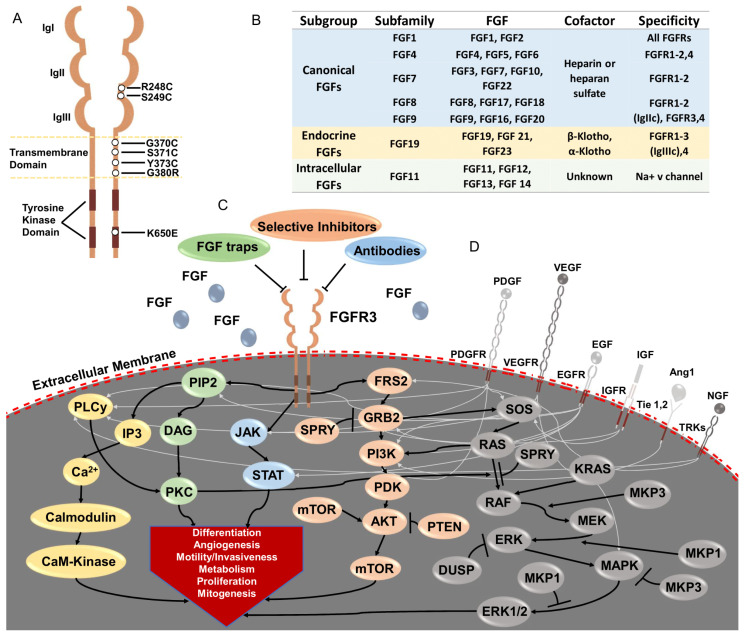
**Overview of FGF/FGFR signaling and its dysregulation in advanced bladder cancer**. (**A**) Basic structure of the fibroblast growth factor receptor (FGFR) family and activating mutations of FGFR3 found in advanced bladder cancer. FGFRs consist of an extracellular domain enclosing three immunoglobulin (Ig)-like domains (IgI, IgII and IgIII), a transmembrane domain, and two tyrosine kinase sub-domains. In advanced bladder cancer, activating missense mutations in *FGFR3* are the dominant genetic alterations in the FGFR family proteins. These mutations are predominantly in the ligand-binding (R248C and S249C) and transmembrane (G370C, S371C and Y373C) domains, Meanwhile, activating mutations in the tyrosine kinase domain (K650E) are less common. (**B**) Twenty-two FGFR ligands in human. FGFs can be categorized into 3 subgroups: (1) canonical FGFs requiring heparin or heparan sulfate as a cofactor to bind FGFR; (2) endocrine FGFs or hormone-like FGFs having low affinity to heparin, and instead, high affinity of α-Klotho and β-Klotho; (3) intracellular FGFs binding to Na+ V channel. Canonical FGFs and endocrine FGFs have different specificities to different FGFRs. (**C**) FGF/FGFR signaling. Ligand binding to an FGFR monomer leads to receptor dimerization and trans-phosphorylation at several tyrosine residues in the intracellular domains of FGFR. This phosphorylation leads to conformational changes within the intracellular domains of FGFR and subsequent recruitment of adapter molecules to initiate signaling events within the cell. Signaling pathways for FGFRs proceed through 5 downstream cascades. Activated FGFRs phosphorylate FRS2, which in turn binds to SH2 domain containing adaptor GRB2. GRB2 then signals through either PI3K /AKT/mTOR or the RAS/RAF/MEK/MAPK cascade after binding SOS. Activated FGFRs can also phosphorylate JAK kinases, which leads to STAT activation. FGFRs can also recruit and phosphorylate PLCγ, thereby initiating signaling through the DAG/PKC or IP3-Ca2+ pathways. These FGFR signaling pathways have critical roles in cell proliferation, differentiation, mobility/invasiveness, metabolism, angiogenesis, and mitogenesis. (**D**) Cross-talk from other tyrosine kinase receptors after binding with their own ligands enable kinase switching and signaling compensation. These receptors include platelet-derived growth factor receptor (PDGFR), vascular endothelial growth factor receptor (VEGFR), epidermal growth factor receptor (EGFR), insulin-like growth factor receptor (IGFR), angiopoietin receptors (Tie1,2) and tropomyosin receptor kinases (TRK). PDGFR and EGFR overlap with all 5 signaling cascades of FGFR. VEGFR also highly overlaps with FGFR, except for JAK/STAT activation. IGFR can activate PI3K/AKT and RAS/MEK/MAPK cascade. TIEL binding by ang1 activates PI3K/AKT signaling. TRK binding by NGF will activate RAS/MEK/MAPK, PI3K/AKT and PLCγ signaling.

**Table 1 cancers-13-04891-t001:** Current combination therapy involving FGFR inhibitors in bladder cancer.

Compounds	Clinical Trial	Clinical Trial ID	Phase	Sponsor
Erdafitinib, Midazolam, Metformin	An Efficacy and Safety Study of Erdafitinib (JNJ-42756493) in Participants with Urothelial Cancer	NCT02365597	II	Janssen Research & Development, LLC (Raritan, NJ, USA)
Derazantinib and Atezolizumab	Derazantinib and Atezolizumab in Patients with Urothelial Cancer (FIDES-02)	NCT04045613	I & II	Basilea Pharmaceutica (Basel, Switzerland)
Futibatinib and Pembrolizumab Combination	Futibatinib and Pembrolizumab Combination in the Treatment of Advanced or Metastatic Urothelial Carcinoma	NCT04601857	II	Taiho Oncology, Inc. (Tokyo, Japan)
Rogaratinib (BAY1163877) in Combination with Atezolizumab	Phase 1b/2 Study of Rogaratinib (BAY1163877) in Combination with Atezolizumab in Urothelial Carcinoma (FORT-2)	NCT03473756	Ib/II	Bayer (Leverkusen, Germany)

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
