# Peer review of "Targetable Pathways in Advanced Bladder Cancer: FGFR Signaling"

_cancers, 2021, doi:10.3390/cancers13194891_

Round 1
Reviewer 1 Report
Xiao et al provide a review of the FGFR signaling pathway ini bladder cancer from a therapeutic standpoint. The paper is easy to follow and covers most of the current literature. The focus on FGFR is appropriate given the high rate of FGFR3 alterations in this tumor and the lower rate of genetic changes in FGFR1/2.
There is substantial evidence indicating that FGFR3 behaves in a rather distinct manner compared - at least - to FGFR1 and the way the paper is written this is somewhat diluted. The mix of comments on the various FGFRs renders interpretation sometimes a bit confusing and I think that making a point about the quite distinct biology of FGFR3 is worthwhile.
Minor comments: there are a few typos in the manuscript.
Reviewer 2 Report
In the current review article, authors have nicely discussed the role of FGFR and combination therapies in advance bladder cancer. I would only suggest to the authors please carefully check some missing references.
Author Response
REVIEWER 2
In the current review article, authors have nicely discussed the role of FGFR and combination therapies in advance bladder cancer. I would only suggest to the authors please carefully check some missing references.
Checked
Reviewer 3 Report
In this manuscript, the authors provide an analysis of the targetable pathwys in advanced bladder cancer with a focus on the FGFR signaling pathway. They review the preclinical and clinical studies on FGFR inhibitors and drug resistance mechanisms to these inhibitors. They also report and discuss the use of combination of FGFR inhibitors with other therapies.
The review is well-written, very clear, very interesting and exhaustive.
I only have minor comments:
1- lines 103-104 p3 "the missense mutations of FGFR3 are associated with FGFR3 mRNA expression in bladder cancer: Does this higher FGFR3 mRNA expression translate in higher expression at the protein level?
2- lines 237-238 p6 The authors mention the following combination therapies: erdafitinib with midazolam and erdafitinib with metformin, what is the rationale behind the use of these combinations?
Author Response
REVIEWER 3
1- lines 103-104 p3 "the missense mutations of FGFR3 are associated with FGFR3 mRNA expression in bladder cancer: Does this higher FGFR3 mRNA expression translate in higher expression at the protein level?
Thanks for the good question. And yes, it is translated in higher expression at protein levels as shown by immunohistochemistry staining. We have included this information and cited the papers (line 115).
2- lines 237-238 p6 The authors mention the following combination therapies: erdafitinib with midazolam and erdafitinib with metformin, what is the rationale behind the use of these combinations?
Like many drugs, midazolam is metabolized in the liver. Dosing patients with erdafitinib and midazolam allows for midazolam to function as a competitive substrate in the liver. This leads to decreased metabolism of erdafitinib, higher serum levels and ultimately improved pharmacokinetics. Midazolam and metformin are substrates for metabolism and clearance of drug in liver and kidney, respectively. Midazolam is a sensitive cytochrome 450 3A (CYP3A) substrate. CYP3A is responsible for the liver metabolism of drugs and therefore either amplify or weaken the action of those drugs. Metformin is a superior substrate for an organic cation transporter 2 (OCT2), a primary renal uptake transporter having major role in the disposition and renal clearance of most drugs (PMID: 16272756). This trial study aims to evaluate the efficacy and safety of the drug combination.